# The Lung in Rheumatoid Arthritis—Friend or Enemy?

**DOI:** 10.3390/ijms25126460

**Published:** 2024-06-12

**Authors:** Maria-Luciana Anton, Anca Cardoneanu, Alexandra Maria Burlui, Ioana Ruxandra Mihai, Patricia Richter, Ioana Bratoiu, Luana Andreea Macovei, Elena Rezus

**Affiliations:** 1Discipline of Rheumatology, Medical Department II, University of Medicine and Pharmacy “Grigore T Popa”, 700115 Iasi, Romania; dr.anton.luciana@gmail.com (M.-L.A.); maria-alexandra.burlui@umfiasi.ro (A.M.B.); ioana-ruxandra_mihai@umfiasi.ro (I.R.M.); patricia.richter@umfiasi.ro (P.R.); ioana.bratoiu@umfiasi.ro (I.B.); luana.macovei@umfiasi.ro (L.A.M.); elena.rezus@umfiasi.ro (E.R.); 2Clinical Rehabilitation Hospital, 700661 Iasi, Romania

**Keywords:** rheumatoid arthritis, interstitial lung disease, RA-ILD, cytokines, ACPA, lung, pathogenesis

## Abstract

Rheumatoid arthritis (RA) is a chronic autoimmune condition frequently found in rheumatological patients that sometimes raises diagnosis and management problems. The pathogenesis of the disease is complex and involves the activation of many cells and intracellular signaling pathways, ultimately leading to the activation of the innate and acquired immune system and producing extensive tissue damage. Along with joint involvement, RA can have numerous extra-articular manifestations (EAMs), among which lung damage, especially interstitial lung disease (ILD), negatively influences the evolution and survival of these patients. Although there are more and more RA-ILD cases, the pathogenesis is incompletely understood. In terms of genetic predisposition, external environmental factors act and subsequently determine the activation of immune system cells such as macrophages, neutrophils, B and T lymphocytes, fibroblasts, and dendritic cells. These, in turn, show the ability to secrete molecules with a proinflammatory role (cytokines, chemokines, growth factors) that will produce important visceral injuries, including pulmonary changes. Currently, there is new evidence that supports the initiation of the systemic immune response at the level of pulmonary mucosa where the citrullination process occurs, whereby the autoantibodies subsequently migrate from the lung to the synovial membrane. The aim of this paper is to provide current data regarding the pathogenesis of RA-associated ILD, starting from environmental triggers and reaching the cellular, humoral, and molecular changes involved in the onset of the disease.

## 1. Introduction

Rheumatoid arthritis (RA) is considered a “state-of-the-art” autoimmune disorder, being defined by a long-standing inflammation that mainly involves the synovial membrane, leading to symmetrical polyarthritis and having complex extra-articular manifestations (EAMs) [1,2,3,4].

After the cardiovascular system, the lungs represent a frequent site for EAMs (the involvement being extensive) from the pleura to blood vessels. Interstitial lung disease (ILD), a multifaceted disease that was first described by Ellman and Ball in 1948, is known as the most important EAM of RA. The rate of survival from the moment of diagnosis is around 2.6 to 3 years [5,6]. The prevalence of RA-ILD is very large, ranging from 6% to 67% and depends on various aspects such as preexisting risk factors, the setup of RA, disease activity, imagistic patterns, and clinical and paraclinical findings [7,8,9,10]. Even if the pulmonary involvement frequently appears after joint destruction, the lungs may become affected many years in advance [3]. The lifetime risk of progressing to RA-ILD is highlighted to be between 6% and 15% [11,12].

An early diagnosis of RA-ILD is very difficult to make due to the asymptomatic stages of the disease. The evolution of ILD may start with some incidental findings of mild diffuse inflammation, swiftly leading up to life-threatening fibrosis. Depending on the radiographic method used, the prevalence of ILD in RA using a lung X-ray can be below 5% and can reach up to 20–30% when high resolution computed tomography (HRCT) is performed [13,14].

Although there are more and more RA-ILD cases, the pathogenesis is incompletely understood [15,16]. Age, male gender, active smoker status, high titers of anticyclic citrullinated peptide antibodies (ACPAs) or rheumatoid factor (RF), genetic predisposition, and high disease activity are risk factors for RA-associated ILD [17,18,19]. As one of the most significant causes of mortality and morbidity, ILD influences the therapeutic decision and interferes with the “treat-to-target” strategy [20].

The exact pathogenetic mechanisms of RA-ILD are still poorly understood. Paulin et al. highlighted the following two possible pathways: (1) the first site of the immune response is located in the synovial tissue, spreading inflammation into the lungs through the Th1, Th2, and Th17 specific cytokine cascade; fibroblasts undergo differentiation into myofibroblasts, which, in turn, generate an important inflammatory status and produce fibrosis in the pulmonary parenchyma (leading to the characteristic appearance of non-specific interstitial pneumonia (NSIP)); (2) the second theory shows a progressive fibrotic lung pattern due to the preexisting immunological and genetic background, generating an aberrant response in the activation of myofibroblasts and alveolar epithelial cells; after the process of citrullination, ACPAs migrate from the lungs to the synovium (usual interstitial pneumonia (UIP)) [21,22]. With regard to the imaging appearance, the UIP pattern is usually highlighted by honeycombing and a subpleural reticular appearance distributed more at the base of the lung. Moreover, the UIP pattern is defined by unusual damage recovery and unusual epithelial cell activity, being an important risk factor for RA-ILD evolution. In addition, the NSIP pattern is also defined by ground-glass opacities with subpleural distribution, occurring in one-third of RA-ILD patients. This pattern is specific for a longer duration of articular symptoms, a decreased risk of disease evolution, and a superior response to treatment compared with the UIP pattern [23]. Thus, new evidence sustains that the response of the immune system is initiated at the level of pulmonary mucosa [6,23,24].

The first piece of evidence in the pathogenesis of RA was published by Erik Waaler in the 1940s; RF playing the most important role at that time. Nowadays, ACPAs are more specific, although their long presence is not always correlated with RA clinical manifestations. RA is an autoimmune disorder in which many types of cells and signaling paths lead to multiorgan damage [25]. Numerous cells, including dendritic cells (DC), subtypes of T and B cells, macrophages, neutrophils, fibroblasts, and osteoclasts, have been involved in the development and maintenance of joint inflammation [26,27]. RA-associated ILD disease is characterized by a complex interplay between environmental triggers and a distinctive genetic background, leading to a specific loss of immune tolerance [28]. The production of RFs and ACPAs is correlated with human leukocyte antigen (HLA)-DRB1 alleles and the PTPN22 variant. Some HLA haplotypes that encode the β-chain of major histocompatibility complex class two (MHC II) antigens, well known as “shared-epitopes”, raise the risk of developing RA [29,30]. 

The aim of this paper is to provide a comprehensive review regarding the pathogenesis of RA-ILD. The literature research included computerized databases such as MEDLINE (PubMed), Web of Science (Clarivate Analytics), and Cochrane Library (Cochrane). Search strategies were assembled using database-specific subject headings and keywords such as rheumatoid arthritis, interstitial lung disease, RA-ILD, cytokines, ACPA, and HLA. The research focused on studies published in the last 10 years (2014–2024). We only selected articles regarding human subject studies. This article is based on data from previously published studies.

## 2. Genetic Predisposition

Of all the genes involved in RA development, HLA-DRB1 alleles (DRB1*01 and DRB1*04; DQ8) represent about 50% [31,32,33]. Different *HLA-DRB* alleles have a certain risk factor in the appearance of RA-ILD, HLA-DRB1*16 and HLA-DRB1*15 being related to a higher risk [34,35]. Mori et al. [36] noticed that RA patients positive for HLA-DRB1*1501 and *1502 alleles had a higher risk of developing ILD. Charles et al. [37] found a correlation between HLA-B40 and lung involvement in RA. They noticed an increased risk of about 40.54-fold for lung involvement compared with other EAMs. Other studies highlighted a higher frequency of HLA-DR4 (60%) and HLA-B54 (63.2%) polymorphisms in RA-ILD patients versus controls [38,39]. Juge et al. [40] highlighted the correlations between RA-ILD and mutations in familial pulmonary fibrosis-linked genes (TERT, RTEL1, PARN, SFTPC). The MUC5B promoter variant could be involved in the occurrence of RA-associated ILD, but it seems that it plays no role in RA pathogenesis. In addition, the MUC5B variant even has a beneficial role in patients with interstitial pneumonia who have RA-ILD or another autoimmune predisposition [6,41,42,43,44,45]. A variant of the MUC5B promoter (rs35705950) is the strongest genetic risk factor for fibrosis and a strong risk factor for RA-ILD, particularly in patients who develop an UIP pattern [46,47,48]. In a recent study, Palomaki et al. showed that MUC5B carriers had a three-fold risk of developing ILD, with men being the most susceptible [49]. In a recent meta-analysis, the variant of RPA3-UMAD1 was included as a risk factor for RA-ILD in the Japanese population [50]. Another study on Japanese subjects presented rs2609255G as an allele associated with a higher risk for developing the UIP pattern in RA-ILD male patients [51].

## 3. Risk Factors for RA-ILD

### 3.1. Age

Advanced age is considered a risk factor in the occurrence of RA-ILD [52,53]. This is due to some changes in the lung parenchyma such as immunosenescence and a decreased regenerative function caused by inflammation and fibrosis [54,55]. RA-ILD occurs frequently in the fifth–sixth decade, while age is considered an independent risk factor for ILD [56,57].

### 3.2. Patient Gender

Regarding the gender of RA-ILD patients, there are various controversial hypotheses. Although male smokers represent a well-defined risk category, there have been studies that reported a higher incidence in women [58]. Male gender has been evidenced as an independent risk factor for RA-ILD, nodules, or bronchiectasis [59,60], and it has been associated with an unfavorable prognostic [61].

### 3.3. Race

In an observational study over a period of 13 years, it has been emphasized that the patient’s gender plays an important role with regard to mortality and morbidity rates. A percentage of 26% of the Hispanic population has been diagnosed with RA-ILD, followed by the White population, and it has been concluded that the Black population has the lowest incidence [62]. Native American patients are more predisposed to developing RA [63]. Indians and Japanese diagnosed with RA also have a higher risk of developing ILD, while severe exacerbations have often been observed in Japanese individuals [64,65].

### 3.4. Smoking

Smoking may be considered the most important exogenous risk factor for RA-ILD, being first described 30 years ago. The risk is correlated with the number of cigarettes smoked and is maintained several years after stopping smoking [66,67]. Various cigarette components have been associated with RA pathogenesis, particularly polycyclic aromatic hydrocarbons (PAHs), which have been shown to trigger a specific transcription factor in synovial DCs, which leads to a higher production of interleukin-6 (IL-6). The ligand-activating transcription factor for PAHs is also present in the synovial tissue of RA patients [68]. Particularly, in RA smokers, joint inflammation is initiated using a subset of synovial DCs, suggesting a possible correlation between DCs, cigarette smoke, and inflammation. Individuals who smoke have been shown to have a higher number of DCs at the level of the airways and interstitial lung tissue [69]. Nicotine can trigger autoimmunity related to RA-ILD pathogenesis due to the secretion of neutrophil extracellular traps (NETs), which are a source of ACPAs and support the activation and secretion of cytokines by synoviocytes [70,71]. Moreover, nicotine can mediate airway and parenchymal lung injury through damage of the endothelial and epithelial cell barriers, which stimulates the recruitment of proinflammatory cells, the secretion of transforming growth factor beta (TGFβ), and the epithelial-to-mesenchymal transition (EMT) [72,73]. Miyake et al. [74] noticed that individuals who smoke have a 2.21 higher risk of developing ILD. Saag et al. [75] reported that smokers with a smoking history of ≥25 pack-years have a 3.8 higher risk for ILD. In another case-control study, Baumgartner et al. [76] demonstrated that people with a history of smoking had a 1.6 higher risk of developing ILD. Restrepo et al. found that patients who had HLA-DRB1 shared epitopes and an active smoker status had an increased susceptibility to develop RA-ILD. This particular HLA genotype increases the risk of developing RA and ACPAs, especially in smokers [17].

### 3.5. External Pollutants

Workplace exposure to silica, ammonium, mineral dust, and black carbon particles increases the risk of developing pulmonary manifestations in RA [77,78,79,80]. Following contact with external pollutants, an elevated number of inflammatory cells and cytokines are secreted and this causes the inflammatory reaction. Regarding the disruption of the immune system, any change participates in the activation of the inflammatory process, both local and systemic. Liu et al. evidenced that environmental pollutants such as PM2.5, PM10, sulfur dioxide, and nitrogen dioxide may be considered causative factors for frequent hospitalizations in RA-ILD patients [81]. The exact mechanism of the involvement of air pollutants in the development of RA-ILD is not yet fully understood.

### 3.6. Disease Activity of RA

Low disease activity or complete remission have been associated with better RA prognosis [82,83]. Using the disease activity score 28 (DAS28) and the clinical disease activity index (CDAI), the activity of RA is the most important risk factor for RA-ILD [84,85]. In a recent study, Zhuo et al. showed that RA may present increased disease activity during ILD progression [86]. RA-ILD female patients in the fifth decade are most likely to have a moderate or severe disease [87]. Sparks et al. highlighted that patients with mild-to-severe disease activity had a double risk for RA-ILD development. By stopping the inflammation due to RA, pulmonary complications can be avoided [84]. Patients who had high activity scores (DAS28, CDAI) also had an increased mortality and morbidity risk due to RA-ILD [85,88,89]. 

### 3.7. Lung Microbiota

Starting in the 19th century, correlations between RA and lung microbiota have been taken into consideration as the “mucosal origin” theory. It was demonstrated that infections with *Porphyromonas gingivalis* (*P. gingivalis)* can induce an immunological response through citrullinated peptides [11,90]. The process is activated by PADs, with arginine components of self-proteins being transformed into neutral citrulline residues. Following these changes, the susceptibility of self-citrullinated proteins to degradation is higher, and many neoepitopes are generated [31]. Moreover, Epstein–Barr virus and bacterial infection with *Proteus mirabilis* and *Escherichia coli* were considered triggers for RA through a molecular mimicry mechanism [26,27,91]. Known as “not sterile” sites, the tracheobronchial structure and the lung parenchyma have complex and diverse microbiomes, which can be easily influenced by triggers such as smoking, antibiotics, exogenous substances, or other endogenous factors. 

In RA patients, microbiota dysbiosis begins at the level of oral cavity and later extends to the pulmonary level [92,93,94]. It was highlighted that early-onset RA was associated with periodontal diseases. Infections with *Prevotella* species or *P. gingivalis* are trigger factors for RA, usually in ACPA-positive patients [95,96,97]. The reactions between bacteria and lung parenchyma influence the innate and adaptive immune system and induce autoimmunity via molecular mimicry, bacteria-induced autoantigen production, and immune regulatory reactions [98,99,100]. Moreover, some studies revealed that ACPAs overreact with citrullinated molecules, which are derived from bacterial species [66,101]. Figure 1 summarizes the main risk factors involved in the development of RA-ILD disease.

## 4. Immunological Profile in RA-ILD

### 4.1. ACPA and RF

Antibodies related to RA, RF, and ACPA (also including carbamylated proteins) appear several years before the clinical manifestations and participate in the development of the disease [45]. The agglutinating IgM RFs confer RA seropositivity [102]. Citrullination represents the post-translational change of peptides, catalyzed by PAD, leading to the transformation of arginine into citrulline. This structural adjustment modifies the intercommunication of the immune system with the proteins. Following this modification, the immune system does not recognize the protein anymore, and therefore auto anticitrullinated antibodies are formed [15]. These autoantibodies can bind to residues of citrullinated protein and “self” proteins like fibrinogen, type II collagen, vimentin, α-enolase, fibronectin, and histones. ACPAs can be found in 60–80% of RA patients, with a specificity up to 96–99%. RF and ACPA-positive patients have a 40% risk of disease onset [103]. Regarding lung damage, ACPAs were found in higher titers in RA-ILD patient serum than in the serum of patients with RA without ILD [104].

The pathogenic role of ACPAs in RA-ILD has not been completely understood, even though affected patients showing higher ACPA serum titers and increased ACPA levels have been correlated with ILD and airway disease. ACPAs can contribute to cell injury through the formation of immune complexes and by increasing the secretion of proinflammatory cytokines like tumor necrosis factor (TNF), IL-6, and IL-8; moreover, ACPAs can increase the secretion of NETs. Along with RF, ACPAs create immune complexes and activate macrophages, increasing the secretion of inflammatory cytokines such as IL-6 and TNF-α [3]. It is interesting that, in pre-RA, ACPAs can be formed up to 10 years before the onset of the first symptoms of the disease. There are studies that have shown an association between ACPAs and the risk of developing bone erosions. In this case, ACPAs may increase bone resorption via the following two ways: (i) direct recognition of citrullinated proteins from the surface of osteoclast precursor cells, which leads to the formation of osteoclasts, or (ii) activation of macrophages mediated by immune complexes, which may release proinflammatory cytokines, which in turn activate osteoclasts. Furthermore, activation and differentiation of IL-8-dependent osteoclasts by ACPAs has been associated with joint pain [105,106]. In addition, ACPA may regulate the release of NETs and neutrophil cell death, thus maintaining inflammation and autoimmunity [34].

Some studies revealed a strong correlation between high titers of ACPAs and ILD regarding the development of RA-ILD [107,108,109]. Dai et al. and Correia et al. showed the role of elevated levels of ACPAs in the higher prevalence of ILD [110,111]. Other studies showed that serum RF and ACPA levels were notably high in patients with RA-ILD [13,39]. Giles et al. indicated that, regardless of the type of ACPAs, the patients showed radiographic lung changes of ILD [112]. In a meta-analysis, Zhu et al. also revealed that the positivity of ACPAs was strongly correlated with a risk of developing RA-ILD [113].

Moreover, antibodies such as IgA and IgG RF can be present in the sputum of patients who have a higher risk of RA (like a first-degree relative with RA), even if RF or ACPAs are not highlighted in the serum. Due to the damaged integrity of the mucosal barrier, these antibodies can later leak into the systemic circulation [114]. Upon lung tissue examination, almost half of RA-ILD patients showed evidence of citrullinated proteins [115].

The percentage of citrullination in idiopathic pulmonary fibrosis is also important and almost identical to that in RA-ILD, which reinforces the essential pathogenic role of citrullination in the occurrence of lung damage [115,116]. Recent studies showed that higher titers of RF in association with smoking can lead to the emergence of RA-ILD [60,61,117]. Moreover, high levels of IgA-RF may be considered an important predictor for RA-ILD evolution [61]. A recent study found that there were other antibodies correlated with the development of ILD in RA patients, such as antibodies against citrullinated alpha-enolasepeptide 1, especially in Chinese and Italian populations [90].

### 4.2. Peptidyl Arginine Deiminases Enzymes

PADs have a decisive role in the process of citrullination. Anti-PAD2 antibodies are correlated with moderate disease and few EAMs, while anti-PAD4 antibodies are associated with a severe and rapidly progressive disease. Anti-PAD3/4XR are reactive antibodies that may signal the development of RA-ILD, especially in the case of non-smoking patients [42,118,119,120]. In a recent cross-sectional study, Nava-Quiroz et al. highlighted that there were some nucleotide variants of PADI2 and PADI4 that were highly correlated with RA-ILD. The rs1005753-GG of PADI2 and rs11203366-AA and rs11203367-GG of PADI4 were strongly connected to genetic predisposition in RA-ILD patients. PAD4 serum titers were elevated in RA-ILD subjects, while increased levels of PAD2 were observed in patients without lung damage [121].

### 4.3. Anticarbamylated Protein Antibodies

Anticarbamylated protein (Anti-CarPs) antibodies (highlighted for the first time in 2011) were correlated with poor prognosis and increased morbidity and were associated with RA-ILD [90,122]. Moreover, Anti-CarP antibodies have been evidenced in patients with other pulmonary pathologies, without RA [42,123,124]. Few Anti-CarP antibodies were in higher serum titers and strongly related to RA-ILD development, e.g., IgG anti-fetal calf serum (FCS), antichimeric fibrin/filaggrin homocitrullinated peptide (Anti-CFFHP), antifibrinogen (Anti-Fib), and IgA anti-FCS [90]. In a cross-sectional study, Castellanos-Moreira et al. showed increased levels of Anti-Fib and Anti-CFFHP and an independent association between ILD and Anti-CFFHP, Anti-FCS, and IgA Anti-FCS. Furthermore, increased levels of RFs and ACPAs were found among patients with Anti-CFFHP, Anti-FCS, IgA Anti-FCS, and Anti-Fib antibodies [122].

### 4.4. Antibodies against Malondialdehyde

Antibodies against malondialdehyde (anti-MMA) are strongly expressed in the lung parenchyma of RA-ILD patients, especially IgA and IgM isotypes. A moderate to severe disease activity was associated with the positivity of anti-MMA antibodies [125,126,127,128].

## 5. Cell Profiles in RA-ILD

### 5.1. Macrophages

Macrophages coordinate the inflammation process and mediate fibrosis formation. These cells are very versatile and show high plasticity and various immunity functions. Macrophages initiate the inflammatory response in the case of an injury and contribute to the repair and resolution of the injury. In the case of the lung and airway microenvironment, macrophages control the development and establishment of ILD. Macrophages and their products are involved in each main step of this process. The appearance of fibrosis occurs through three mechanisms. First, is the reduction of PAD2 in fibroblasts and the reduction of collagen deposits in the lungs; secondly, macrophages secrete high levels of IL-6, which determine the proliferation of T cells; the third stage involves high levels of TNF-α produced by macrophages that lead to inflammation and fibrosis in the lungs [129]. Two types of macrophages are defined regarding inflammation and immune modulation: M1 (classical or proinflammatory macrophage) and M2 (alternatively activated) [68]. M1 macrophages induce myofibroblast degradation and deteriorate the extracellular matrix, being present at the beginning of the injury. On the other side, M2 macrophages turn on fibroblasts over TGF-β1 and platelet-derived growth factor (PDGF-β) and suppress extracellular matrix deterioration, being involved in reducing the inflammatory process [52]. In the pathogenesis of RA-ILD, through the phagocytic function acquired during the inflammatory process, these two categories of macrophage contribute to the development of fibrosis in the lung parenchyma [130]. This macrophage activation plays an important role by amplifying the cytokine storm, inducing secretion of cytokines, activating the Th1/Th17 pathway, and suppressing regulatory cells [130,131]. M1 macrophages are involved in the process of fibrosis through the secretion of proinflammatory cytokines such as IL-6, IL-12, IL-23, IL-1β, and TNF-α. In addition, M2 macrophages contribute to fibrosis particularly by secreting profibrotic cytokines such as IL-4, IL-10, IL-33, and TGF-β [131]. Macrophage autoimmune–complement–interferon cascade genes are modified in RA-ILD and can have a profibrotic inflammatory response in the lungs. TNF-α leads to an inflammatory phase dominated by cell infiltration, followed by the fibrotic phase characterized by the irreversible deposition of collagen fibers in the pulmonary parenchyma [129,132].

Similar to fibroblasts, subpopulations of macrophages like alveolar macrophages, macrophages derived from bone marrow, and those derived from monocytes are involved in lung fibrosis; depending on the stages of fibrotic diseases, alveolar macrophages can also have antifibrotic effects. In a preclinical study, alveolar cell injury can activate inflammation and macrophages derived from monocytes, leading to fibrosis [133]. Lung samples from patients diagnosed with fibrosis showed various proportions of macrophage populations when compared with normal lungs. The high inflammatory capacity of alveolar macrophages in ILD is suggested by the increased release of reactive oxygen species (RoS), plasminogen activators, mitogens from mesenchymal cells, cytokines, chemokines, leukotrienes, and growth factors [134]. These macrophages have common alveolar and interstitial macrophage characteristics and profibrotic immune cell functions. In an inflamed synovium, the activated macrophages secrete an increased number of cytokines like IL-1β, TNF-α, and IL-6. These cytokines can recruit and activate other natural immune cells such as neutrophils. In turn, the activated neutrophils release high levels of cytokines and oxidants, like phospholipases, myeloperoxidases, proteases, defensins, and TNF-α, favoring joint destruction [135]. On the other hand, proinflammatory cells play a role in fibrosis via interleukin IL-4 and IL-13, inducing the production of TGF-β [136].

### 5.2. Neutrophils

Neutrophils produce tissue inhibitors of metalloproteinases and neutrophil elastase. They activate TGF-β and determine other inflammatory cells to migrate into the lungs, promoting pulmonary fibrosis [137,138].

Neutrophils have the highest cytotoxic potential of all the cells involved in the RA pathogenesis. This is achieved due to their ability to release reactive oxygen species and enzymes with a degrading role. They actively participate in the chemokine and cytokine cascades. Moreover, neutrophils regulate immune responses through cell–cell interactions. There is evidence that suggests that neutrophils play an established role in inflammation due to the release of NETs, which contain chromatin associated with granule enzymes. The neutrophils of RA patients have a higher predisposition to release NETs. They eliminate extracellular microorganisms and represent a source of autoantigens. This role was unrecognized before. Citrullinated proteins, which may act as neoepitopes in the situation of immune tolerance loss, are produced by PADs [39]. In RA patients, antibodies against citrullinated proteins can be found before any clinical symptoms, predicting an erosive disease. Neutrophils play an immunoregulatory and cytotoxic role in RA and may also represent an important source of autoantigens that drive the autoimmune processes [39].

The receptors of neutrophils can make interactions with vascular endothelial cells, which regulate the attachment and later migration of neutrophils from the circulation into tissues during inflammation. These circulating neutrophils are different and have specific properties. Inflammatory neutrophils have functions similar to macrophages, leading to the increased release of inflammatory molecules such as chemokines, cytokines, and MHC class II antigens. Most of these functions are the result of the selective changes in gene expression that take place upon neutrophil activation during inflammation [39].

In general, circulating neutrophils do not have a wide lifespan and experience constitutive apoptosis after 24 h. While migrating from the circulation into inflamed sites, the function and longevity of neutrophils is altered. Apoptotic neutrophils are inert in terms of function due to programmed shutdown of their receptor signal transduction pathways. Neutrophils express receptors on cell surfaces, allowing them to be recognized, and undergo phagocytosis by macrophages of other phagocytic cells. At inflammatory sites, this removal of apoptotic neutrophils can stop tissue damage. In the case of inflammatory diseases, impaired neutrophil apoptosis determines a longer survival, which leads to an increased release of cytotoxic products, immunoregulatory chemokines, and cytokines, resulting in prolonged inflammation. Hypoxia and antiapoptotic cytokines like TNF and IL-8 can extend the survival of neutrophils up to several days [6,39].

### 5.3. Fibroblasts

Fibroblasts are also very important because they offer the initial scaffold for the generation of tissue in wound healing [139]. Fibroblasts can differentiate into myofibroblasts, which support the fibrotic process, after they gain a profibrotic phenotype resistant to apoptosis [35]. This process is considered to have a crucial role in fibrosis pathogenesis, being defined as fibroblast-to-myofibroblast transition (FMT) [40,140]. The senescence of lung parenchyma cellularity plays a key role in pulmonary damage by activating FMT [141,142]. In this process, both adaptive and innate immune cells are involved. Fibrocytes produce chemokines, growth factors, matrix metalloproteinases (MMPs), and cross-linking enzymes, and promote pulmonary fibrosis [137,139]. There is another process called EMT, in which epithelial cells gain mesenchymal characteristics instead of epithelial characteristics. This transformation leads to deposits of myofibroblasts, which are correlated with ILD progression. Both endothelial-to-mesenchymal transformation and EMT represent other sources of myofibroblasts in the pathogenesis of RA-ILD, taking place in the presence of TGF-β. These two transitions involve the following four significant steps: decrease in epithelial adhesion properties, structural reorganization, elevated permeability, and higher capacity of migration [143,144]. Both transitions to mesenchymal transformation contribute to the pathogenesis of RA-ILD, favoring the fibrotic process by maintaining inflammation [130]. Figure 2 illustrates the role of macrophages, neutrophils, and fibroblasts in the pathogenesis of RA-ILD.

### 5.4. Dendritic Cells

DCs play a crucial role in cellular immune reactions by absorbing, processing, and presenting antigens to T and B cells. The DC phenotype, which expresses surface molecules and produces chemokines and cytokines, controls the equilibrium between the activation of the immune system or the induction and preservation of immune tolerance. There is increasing evidence that a changed function and distribution of DCs in RA-ILD favors autoimmune inflammation [35]. Regarding this, a reduced frequency of plasmacytoid DCs, and the presence of conventional DCs in the plasma of patients with RA, was reported to induce an increased DC migration into inflamed joints [145]. At the level of the joints, DCs produce an increased secretion of IL-23 and IL-12, which support Th17 responses, leading to an imbalance between Th2, Th1, and Th17 responses. Inflammatory DCs from the synovial fluid play an important role in the pathogenesis of RA-ILD by activating Th17 cells through the production of IL-1β, IL-23, IL-6, and TGF-β. In addition, activated plasmacytoid dendritic cells (pDCs) contribute to systemic inflammation through the secretion of interferon (IFN)-β, IFN-α, IL-23, and IL-18. Patients with RA-ILD and ACPAs have an increased level of pDCs in the synovial membrane compared with ACPA-negative patients [4,145].

### 5.5. T and B Cells

Both T and B cells are very important in the inflammatory process. T cells participate in the activation and formation of myofibroblasts through IL-17 and IL-13. The fibrotic modifications in the lung parenchyma are related to a disturbed Th1/Th2 balance. Th1 is considered the suppressor of fibrosis, while Th2 favors this process. B cells are correlated with ACPA formation, thus inducing the production of IL-6, IL-8, and TNF [35]. T-lymphocytes support the differentiation of B cells after exposure to specific antigens [90]. Studies highlighted that CD4^+^ and CD8^+^ T cells were more involved in the development of RA-ILD, particularly in smoking patients [11,44,146,147].

## 6. Cytokine Profile in RA-ILD

Inflammation of the alveolar epithelium can be induced by many environmental factors like oxidative stress, pollutants, infection, or smoking [34,148,149]. In general, inflammation triggers the repair pathway that includes normal fibroblasts and myofibroblasts, which leads to a normal homeostatic response that heals and restores the lung anatomy [150]. Proinflammatory cytokines influence important biological processes such as cell growth, differentiation, proliferation, regulation of the immune response, and tissue repair [151,152]. This pathway mediated by cytokines is crucial for RA pathogenesis [153,154].

### 6.1. TNF-α

TNF-α is one of the main regulators. TNF-α is a proinflammatory cytokine associated with ILD evolution. It stimulates the proliferation of fibroblasts and activates the expression of cytokines, PDGF-β, chemokines, and TGF-β [34,40,139]. TNF-α inhibitors are frequently used in the treatment of RA. However, more and more studies have suggested that TNF-α inhibitors are correlated with ILD development and can cause pulmonary toxicity [155,156]. Cytokines secreted by T cells are IFN-γ, interleukin (IL)-17A, receptor activator of nuclear factor KB ligand (RANK-L), and TNF-α [27]. TNF-α is also produced by B cells, NK-cells, and synovial macrophages. TNF-α has been proven to induce bone resorption and cartilage degradation; it also increases RANK-L secretion by osteocytes [157]. There are studies that have highlighted that TNF-α can directly stimulate the differentiation of the monocyte/macrophage lineage cells into osteoclasts through a mechanism independent of RANK-L. The second key role of TNF-α in inflammation is the capacity to stimulate the production of other inflammatory cytokines like IL-6 and IL-1β. These cytokines attract leukocytes and promote the formation of an inflammatory milieu in the synovial membrane [4].

Regarding other cytokines, IL-4 and IL-18 are also elevated in the serum of patients with ILD [42,158]. In addition, IL-13 is high, being associated with lung fibrosis in HRCT [159,160,161,162,163]. IL-11 and IL-33 are correlated with RA-ILD [162,163]. Zhang et al. highlighted that lung fibroblast expressed numerous IL-17A receptors, while IL-23 was associated with the EMT process in RA-ILD [164,165]. Table 1 synthesizes the main interleukins involved in the development of RA-ILD.

### 6.2. IL-17

IL-17 participates in the activation of osteoclastogenesis and of fibroblast-like synoviocytes and in the recruitment and activation of B cells, macrophages, and neutrophils [166]. Of the six members of the IL-17 cytokine family (IL-17A to IL-17F), IL-17A was the first one described [151]. IL-17A produced by Th17 cells favors neutrophil recruitment and the formation of other proinflammatory cytokines (IL-8 and IL-6 by fibroblastic, epithelial, and endothelial cells). Through all these processes, IL-17A contributes to neoangiogenesis, cartilage destruction, and bone erosion in patients with RA [33,167]. IL-17A has also been shown to increase the production of MMP-1 by synoviocytes, leading to cartilage destruction [4].

IL-17 has two main roles: firstly, to promote and initiate chemotaxis, and secondly to recruit and activate neutrophils into inflamed tissues [168]. IL-17 interacts with the TGF-β signaling pathway to induce pulmonary fibrosis [169].

Figure 3 illustrates the main role of IL-17 and TNF-α in RA-ILD.

Other molecules related to RA-ILD pathogenesis are MMPs. Resulting from epithelial degradation, these proteases play an important role in EMT and are involved in extracellular matrix degradation [90,170]. Moreover, MMPs sustain the connections between inflammatory and fibrotic processes by activating T and B cells, macrophages, and neutrophils [90]. The EMT fibrotic process is maintained by MMP-3, and these proteases are involved in RA-ILD pathogenesis [171,172]. MMP-7 can be considered a biomarker for RA-ILD, while high levels of MMP-3 and MMP-7 are associated with RA activity [173,174,175,176]. Chen et al. highlighted increased titers of MMP-7 in RA-ILD patients [137]. Doyle et al. tried to diagnose ILD in patients with RA using biomarkers such as MMP-7, activation/regulation chemokines, and surfactant protein D [175]. One study found another maker strongly associated with the development of RA-ILD, namely LOXL2 [177,178].

Growth factors, other than cytokines and chemokines, take part in the process of fibroblast transformation, forming a bridge between inflammation and fibrosis [90]. PDGF-β is a molecule that has both a profibrotic and a proinflammatory role, being secreted by macrophages, fibroblasts, and epithelial and endothelial cells. In the pathogenesis of ILD, PDGF-β is as important as TNF-α and TGF-β. The development of lung fibrosis depends on the transformation of fibroblasts into myofibroblasts. Regarding chemokines, their contribution is not well established in the lung inflammatory process [90].

The Janus kinase (JAK)/signal transducer and activator of transcription (STAT) is a significant process generated by growth factors. This favors the transformation of macrophages into an inflammatory phenotype, leading to the secretion of IL-6 and TNF-α and promoting fibrosis. Cytokines that play important roles in the development of ILD such as IL-4, IL-13, IL-11, and IL-3 also stimulate the JAK/STAT pathway [179,180,181].

Krebs von den Lungen 6/MUC1 (KL-6) is a mucin-like glycoprotein that is involved in the fibrotic process, maintaining the inhibition of fibroblast apoptosis in the lungs. KL-6 is activated by type II pneumocytes and by epithelial cells from lung parenchyma. After the initiation of lung damage, this protein is considered a key sign for epithelial damage [182,183]. Being evidenced in increased titers in the serum of RA-ILD patients, KL-6 may contribute to the early detection of ILD [42,182]. It is also correlated with higher mortality and morbidity [184]. Lee et al. evidenced correlations between a certain lung pattern in HRCT, disease activity, and high titers of KL-6 in patient serum with RA-ILD [182].

**Table 1 ijms-25-06460-t001:** Interleukins associated with RA-ILD.

Interleukins	References	Pathogenic or Clinical Correlations
IL-4	[161]	↑ RA-ILD
IL-18	[163]	↑ RA-ILD
IL-13	[159,160,161,162,163]	↑ RA-ILD; lung fibrosis on HRCT
IL-11	[160,161]	Correlated with RA-ILD activity
IL-33	[160,161]	Correlated with RA-ILD activity
IL-23	[164,165]	Associated with EMT
MMP-3	[173,174,175,176]	Correlated with RA-ILD activity
MMP-7	[173,174,175,176]	Correlated with RA-ILD activity
PDGF-β	[90]	Profibrotic/inflammatory role
JAK/STAT	[179,180,181]	Associated with lung fibrosis
KL-6	[182,183,184]	↑ RA-ILD; lung fibrosis on HRCT

IL—interleukin, HRCT—high resolution computed tomography, EMT—epithelial-to-mesenchymal transition, ↑—increased.

## 7. Conclusions

Nowadays, RA is nominated as a “decades-long” disease, and it stands out for its complex and distinguished immunological and genetic background, heterogeneity, and the ability to develop complications in time. RA-ILD is a severe and uncertain global problem, whose pathogenesis is not yet fully understood; the mystery remains whether the onset belongs to the inflammatory process located in the joints or whether it is a consequence of pulmonary parenchyma damage. ILD is known as a serious EAM, the course of RA-ILD being associated with a triple risk for morbidity and mortality.

Many new horizons are being revealed regarding the pathogenic pathways in RA-ILD. It is very important to take into consideration each potential trigger that may influence the development and progression of clinical manifestations. Following the causal relationship between RA and ILD, there is a certain mutual influence. Regarding a serious double-meaning problem, RA affects the lung parenchyma through the autoimmune disorder. In addition, new theories regarding the citrullination process in the lung target this aspect as an extremely important trigger for RA.

The activation of the innate and acquired immune system, the loss of self-tolerance, the activation of numerous cells, and the production of an inflammatory environment that causes tissue destruction remain the key points of these diseases.

## Figures and Tables

**Figure 1 ijms-25-06460-f001:**
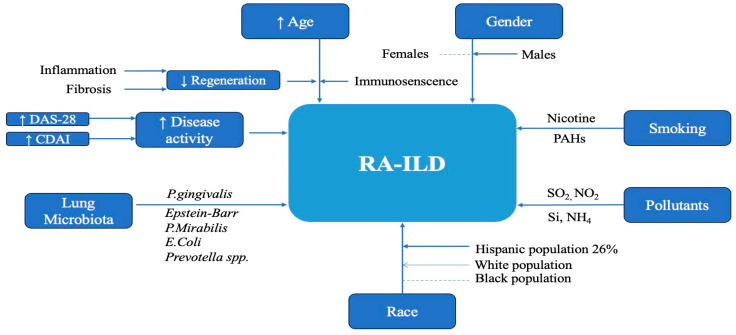
Risk factors associated with RA-ILD. SO_2_—sulfur dioxide, NO_2_—nitrogen dioxide, Si—silica, NH_4_—ammonium, PAHs—polycyclic aromatic hydrocarbons, DAS-28—disease activity score in 28 joints, CDAI—clinical disease activity index.

**Figure 2 ijms-25-06460-f002:**
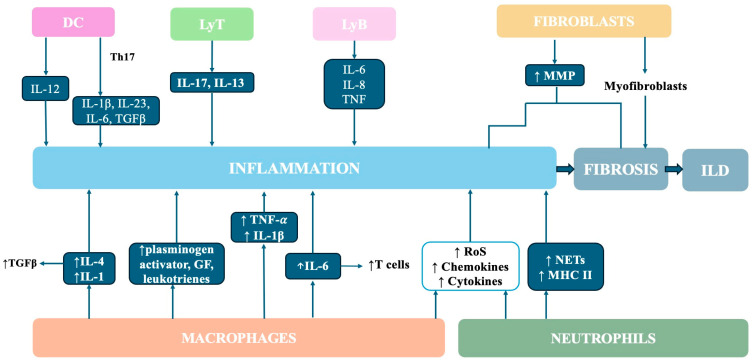
The role of macrophages, neutrophils, fibroblasts, dendritic cells, and T and B cells in the pathogenesis of RA-ILD. MMP—matrix metalloproteinase, TGFβ—transforming growth factor beta, IL—interleukin, GF—growth factor, TNF-α—tumor necrosis factor alpha, RoS—reactive oxygen species, NET—neutrophil extracellular trap, MHC II—major histocompatibility complex class II, DC—dendritic cells, LyT—lymphocyte T, LyB—lymphocyte B.

**Figure 3 ijms-25-06460-f003:**
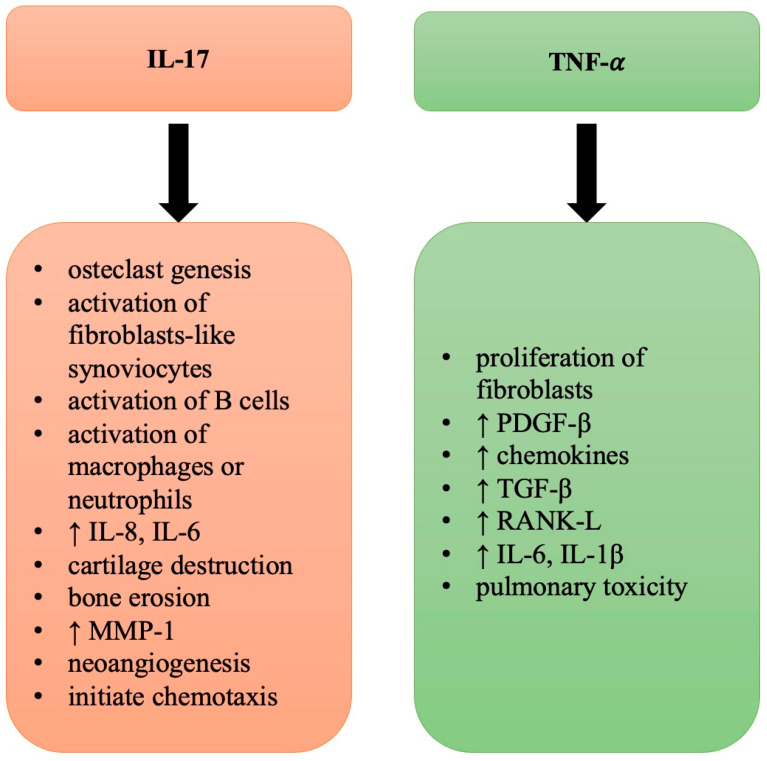
The role of IL-17 and TNF-α in RA-ILD. IL—interleukin, MMP—matrix metalloproteinase, PDGF-β—platelet-derived growth factor beta, TGF-β—transforming growth factor beta, RANK-L—receptor activator of nuclear factor KB ligand, ↑—increased.

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
