# Peer review of "The Lung in Rheumatoid Arthritis—Friend or Enemy?"

_ijms, 2024, doi:10.3390/ijms25126460_

Round 1
Reviewer 1 Report
Comments and Suggestions for Authors
-The authors present a review of RA-ILD pathogenesis
-They should review the grammar throughout the manuscript, particularly in the section 3.1 Age, 3.3 Race,
-In Section 4.1. ACPA and RF, what do they mean by some phrases like:
“Regarding lung damage, ACPAs were found in higher titers in RA-iLD patient serum”. Higher than in which other population?
“Moreover, antibodies such as IgA and IgG RA can be present in the sputum of patients who have a higher risk of RA (like first-degree relative with RA), even if RF or ACPAs are not highlighted in the serum”. What do they mean by IgA and IgG RA?
-Also, third and fourth paragraph of that same section are repetitive and should be restructured and summarized.
In the macrophages section please discuss the M1 and M2 subtypes and their role in inflammation and immune-modulation.
Please discuss endothelial to mesenchimal transition and epithelial to mesenchimal transition role in the pathogenesis of RA-ILD. This is central information that they should include.
In the next sections regarding several cell subtypes and cytokines, the information that they present pertains either to general functions of the cell types and cytokines or to their function in RA in general. Please discuss information pertinent to RA-ILD, not general functions nor their role solely in RA, and please correlate them in a general ILD scheme.
Also please discuss which are the differences in the pathology and pathogenesis of NSIP and UIP patterns of RA associaated ILD.
Please include figures with higher quality and it would be very illustrative to include all of the involved cellular types and cytokines into a main figure.
Comments on the Quality of English LanguagePlease review some grammar errors like plural and singular concordance and send to review the manuscript to a native English speaker or an experienced writter.
Author Response
Response
Dear Reviewer,
Thank you for taking the time to review this paper. We believe that your feedback enhances the scientific quality of this article and contributes to a more in-depth understanding.
In accordance with your recommendations, we have made the following modifications:
- We reviewed the grammar particularly in the section 3.1 Age and 3.3 Race.
- We made the changes required in section 4.1 regarding ACPA and RF.
- We restructured and summarized paragraphs 3 and 4 of the same section.
- We added a paragraph regarding M1 and M2 macrophages and also, we include in discussion their role in inflammation and immune modulation.
- We added a paragraph regarding endothelial to mesenchymal and epithelial to mesenchymal transition and their role in the pathogenesis of RA-ILD.
- We have attempted to include in the paper the current data regarding the cells and cytokines involved in the manifestation of pulmonary symptoms in RA, and we have modified Figure 2. to include other cells (DC,LyB,LyT), as well as the main pro-inflammatory cytokines secreted by these cells.
- We modified all figures, and we included the ones with a higher quality.
- We added a paragraph regarding NSIP and UIP specific lung pattern.
We hope these adjustments meet your approval. We look forward to your response at your earliest convenience and any further suggestions you may have.
Best regards,
The Authors.
Reviewer 2 Report
Comments and Suggestions for Authors
It was a pleasure to review this paper. The authors should be complimented for all their efforts,
The abstract is clear and concise and provides an excellent scope of the objectives
The risk factors section and immunological profile section is well-organised and each factor is discussed in detail. The mechanism linking external pollutants to RA ass-ILD is lacking detail and the authors should consider to elaborate this further.
Has the authors considered including epigenetics as a part of the mechanisms of pathogenesis of RA-ass ILD?
Was it the intention not to include any information on the clinical presentation, radiological findings and treatment of RA-ass ILD? This is just a query and not necessarily a critique?
Comments on the Quality of English Language
The language in its current form (the entire paper) is not acceptable, some sentences are not scientifically sound, some are too long and can be clearer and more concise. Some of the abbreviations are lacking definitions or explanations.
Author Response
Response
Dear Reviewer,
Thank you for taking the time to review this paper. We believe that your feedback enhances the scientific quality of this article and contributes to a more in-depth understanding.
In accordance with your recommendations, we have made the following modifications:
- We added a paragraph regarding the mechanism involving the external pollutants in the development of RA-ILD.
- Unfortunately, due to the very limited data on epigenetics, we did not include this subject in the paper. However, if you consider it to be of significant importance, we can search for and add some information on this topic.
- This paper focuses solely on the presentation of the etiopathogenetic mechanisms of RA-ILD, without addressing the clinical manifestations, radiographic findings, or treatment. If you deem it appropriate, we can modify the title of the paper to specify that it only presents data on etiopathogenesis.
- We revised the grammatical errors and we tried to make sentences clearer and more concise.
We hope these adjustments meet your approval. We loof forward to your response at your earliest convenience and any further suggestions you may have.
Best regards,
The Authors.
Reviewer 3 Report
Comments and Suggestions for Authors
The review addresses rheumatoid arthritis associated interstitial lung disease. Several reviews have been recently published in the general area and this adds to this body. There is an overview of disease burden, epidemiology and prognosis in the introduction, which identifies the issues and overall hypotheses. The remainder of the article discusses the science relating to the pathogenesis of the disorder. This takes a reductionist approach breaking the literature down into data relating to risk factors, cell types and individual cytokines/immunomodulators. There is wide coverage of the literature throughout these sections. The review however would benefit from a critical analysis of what overall conclusion can be drawn on the links between RA and lung disease, the title sets up a question and to a large extent this is not answered in the conclusion.
The standard of English needs to be reviewed to improve grammar, flow and remove typographical errors
Eg line 45 “ILD may starts”, should be ILD may start
Line 63 “immunological and genetical”, should be immunological and genetic.
Line 73 “In the development and maintenance of joint inflammation are implicated connections between numerous cells: dendritic cells”. Needs to be re-drafted perhaps “Numerous cells including dendritic cells have been implicated in the development and maintenance of joint inflammation.
Line 127, the sentence highlights gender but the paragraph is detailing race.
Comments on the Quality of English LanguageThe standard of English needs to be reviewed to improve grammar, flow and remove typographical errors
Eg line 45 “ILD may starts”, should be ILD may start
Line 63 “immunological and genetical”, should be immunological and genetic.
Line 73 “In the development and maintenance of joint inflammation are implicated connections between numerous cells: dendritic cells”. Needs to be re-drafted perhaps “Numerous cells including dendritic cells have been implicated in the development and maintenance of joint inflammation.
Line 127, the sentence highlights gender but the paragraph is detailing race.
Author Response
Respons
Dear Reviewer,
Thank you for taking the time to review this paper. We believe that your feedback enhances the scientific quality of this article and contributes to a more in-depth understanding.
In accordance with your recommendations, we have made the following modifications:
- Although the title is broad, the paper focuses solely on the pathogenic mechanisms involved in the development of RA-ILD. If you believe it is appropriate, we can modify the title of the paper to specify that it only presents data on etiopathogenesis.
- We added a paragraph in the conclusion section which tries to answer the question from the title.
- We revised the grammatical errors according to recommendations.
We hope these adjustments meet your approval. We loof forward to your response at your earliest convenience and any further suggestions you may have.
Best regards,
The Authors.
Round 2
Reviewer 1 Report
Comments and Suggestions for Authors
I previously suggested to discuss M1 and M2 macrophages and endothelial to mesenchymal transition as well as epithelial to mesenchymal transition in the pathogenesis of RA-ILD.
You only included the definitions of M1 and M2 macrophages and the definition of entoMT and EMT into the text but you did not include the references and the information pertaining to the participation of these cells and these processes in RA-ILD pathogenesis. Please include it.
Comments on the Quality of English Language
Please send your manuscript to a native English speaker editor, there are many grammatical mistakes that make it difficult to read the manuscript. I see that you made minor corrections, but the article still needs improving.
Author Response
Dear Reviewer,
Thank you for taking the time to review this paper. We believe that your feedback enhances the scientific quality of this article and contributes to a more in-depth understanding.
In accordance with your recommendations, we have made the following modifications:
- We reviewed the grammar in the entire article.
- We added a paragraph regarding M1 and M2 macrophages and we include in discussion their role in the pathogenesis of RA-ILD.
- We added a paragraph regarding endothelial to mesenchymal and epithelial to mesenchymal transition and their role in the pathogenesis of RA-ILD.
- We added two new references in the bibliography regarding the paragraphs about macrophages, EMT and endoMT.
We hope these adjustments meet your approval. We look forward to your response at your earliest convenience and any further suggestions you may have.
Best regards,
The Authors.